# Associations between Allergic and Autoimmune Diseases with Autism Spectrum Disorder and Attention-Deficit/Hyperactivity Disorder within Families: A Population-Based Cohort Study

**DOI:** 10.3390/ijerph19084503

**Published:** 2022-04-08

**Authors:** Dian-Jeng Li, Ching-Shu Tsai, Ray C. Hsiao, Yi-Lung Chen, Cheng-Fang Yen

**Affiliations:** 1Department of Addiction Science, Kaohsiung Municipal Kai-Syuan Psychiatric Hospital, Kaohsiung 80276, Taiwan; u108800004@kmu.edu.tw; 2Department of Nursing, Meiho University, Pingtung 91200, Taiwan; 3Graduate Institute of Medicine, College of Medicine, Kaohsiung Medical University, Kaohsiung 80708, Taiwan; 4Department of Child and Adolescent Psychiatry, Chang Gung Memorial Hospital, Kaohsiung Medical Center, Kaohsiung 83301, Taiwan; jingshu@cgmh.org.tw; 5School of Medicine, Chang Gung University, Taoyuan 33302, Taiwan; 6Department of Psychiatry and Behavioral Sciences, University of Washington School of Medicine, Seattle, WA 98105, USA; rhsiao@u.washington.edu; 7Department of Psychiatry, Children’s Hospital and Regional Medical Center, Seattle, WA 98105, USA; 8Department of Psychology, Asia University, Taichung 41354, Taiwan; 9Department of Healthcare Administration, Asia University, Taichung 41354, Taiwan; 10Department of Psychiatry, School of Medicine, College of Medicine, Kaohsiung Medical University, Kaohsiung 80708, Taiwan; 11Department of Psychiatry, Kaohsiung Medical University Hospital, Kaohsiung 80756, Taiwan; 12College of Professional Studies, National Pingtung University of Science and Technology, Pingtung 91201, Taiwan

**Keywords:** ADHD, ASD, allergic disease, autoimmune disease, psychological well-being

## Abstract

Autism spectrum disorder (ASD) and attention-deficit/hyperactivity disorder (ADHD) are commonly comorbid with allergic and autoimmune diseases in children. The aim of the current study was to investigate the association between children’s and first-degree relatives’ (i.e., mother, father, and full sibling) allergic and autoimmune diseases and children’s ASD and ADHD. We enrolled participants from Taiwan’s Maternal and Child Health Database. We used the Cox regression model to examine the associations of familial, siblings’ and children’s allergic and autoimmune diseases with children’s ASD and/or ADHD. In total, we included 1,386,260 children in the current study. We found the significant association between familial allergic or autoimmune disease and development of ASD or ADHD among children. We also identified the predominant impact of familial aggregation on the above associations. The associations between some parental diagnoses of autoimmune or allergic diseases in children’s ASD and/or ADHD were stronger in mothers than those in fathers. Early assessment of the possibility of ASD and ADHD is required for children who have a parent with an allergic or autoimmune disease.

## 1. Introduction

### 1.1. Etiologies of Autism Spectrum Disorder and Attention-Deficit/Hyperactivity Disorder

Autism spectrum disorder (ASD) and attention-deficit/hyperactivity disorder (ADHD) are both neurodevelopmental disorders [1]. Researchers have placed increasing attention on the etiologies of these disorders, and multidimensional factors contribute to them. Regarding the genetic etiologies, a genome-wide association meta-analysis demonstrates 12 independent loci of genetic variants in ADHD, supporting the heritable characteristics of ADHD [2]. In addition to genetic factors, the environmental factors also play an important role in developing ADHD [3]. Moreover, multidimensional factors, such as gene defects and chromosomal anomalies [4,5] and low birth weight preterm [6,7], contribute to the development of ASD among children. Further etiological research is beneficial in clarifying the complicated etiologies of ASD and ADHD.

### 1.2. Effects of Immunological Diseases on ASD and ADHD

Researchers have become increasingly interested in the effects of immunological diseases, such as allergic and autoimmune diseases, on the development of ASD and ADHD. Research suggested that children with ADHD or ASD have an increased risk of allergic comorbidities [8]. Moreover, a population-based study in Denmark reported that personal and maternal histories of autoimmune diseases are associated with an increased risk of ADHD [9]. Research has also reported a significant association between the increased level of inflammatory cytokines and ADHD and ASD [10,11], as well as the significant association between upregulating T helper (Th) cell-2 and Th cell-17 and ASD [12]. This evidence suggests that the permeation of peripheral immune cells across the blood–brain barrier may alter neural functions and increase the risk of mental disorders, such as ASD and ADHD [13]. Several genetic studies have also revealed the potential association of ADHD with major histocompatibility complex genes, which are involved in the development of autoimmune diseases, such as juvenile rheumatic arthritis and systemic lupus erythematosus (SLE) [14]. Moreover, a recent genome-wide association study also highlighted the genetic association between autoimmune/allergic disease and ASD and ADHD [15].

### 1.3. Aim of Current Study

Although previous studies have explored the link between immunological diseases and ASD and ADHD, no conclusion has been reached regarding the effects of parental allergic and autoimmune diseases on the development of ASD and ADHD in their children. A population-based study in Canada indicated that children born to mothers with SLE had an increased risk of ASD [16]. Another nationwide study in Sweden reported the significant association between patients’ ADHD and autoimmune diseases in their first-degree relatives [17]. However, a population-based study in Taiwan demonstrated that children born to mothers with SLE or rheumatoid arthritis did not exhibit a greater risk of ASD [18]. Second, whether parental, siblings’ and children’s allergic and autoimmune diseases have various or cumulative effects on the development of ASD and ADHD in children have not been examined. In response to these two knowledge gaps, the aim of the current study was to investigate the association between parental, siblings’ and children’s allergic and autoimmune diseases with the development of ASD and ADHD among children.

## 2. Methods

### 2.1. Population

The current study derived data from Taiwan’s National Health Insurance Research Database (TNHIRD). Information in TNHIRD is based on the National Health Insurance Program, with 99.99% coverage of Taiwan’s population. The TNHIRD has a different dataset to store information, including registry for beneficiaries, ambulatory care claims, inpatient claims, prescriptions dispensed at pharmacies, registry for medical facilities, and registry for board-certified specialists [19]. According to the policies of Taiwan’s National Health Insurance (TNHI), medical claims are sent to the Bureau of TNHI for cross-checking and validation on the purpose of ensuring the adequacy of diagnosis coding. Hospitals or clinics that are found to have fraudulent coding, overcharging, or malpractice are subjected to penalties or suppression of the treatment fees. A recent validation study also support the validity of TNHIRD diagnostic codes of mental disorders [20].

We used the Taiwan’s Maternal and Child Health Database (TMCHD), one of datasets from TNHIRD, for the period from 2004 to 2016. TMCHD is a specific dataset, which was designed to examine the family study by further integration of information about newborn information (i.e., parental identity, child identity, and year of childbirth) since 2004 in Taiwan. The TMCHD contains records on 99.78% of all births nationwide in Taiwan since 2004 [21]. We further linked the TMCHD to obtain the ambulatory care and inpatient claims data for their diagnosis.

The inclusion criteria of the current study were that all of the living born children and their parents did not have missing information on their identity in the TMCHD. We recruited children who were 5 years old or older in 2016 because children aged 5 years or older may develop sufficient symptoms to permit reliable ASD and ADHD diagnoses, and every child had a minimum follow-up period of 1 year (to 2017). In Taiwan, children can enter the kindergarten at two years old; teachers have enough time to observe children’s behaviors for clarifying symptoms of ASD or ADHD. Therefore, it is suitable to set 5 years old as one of the inclusion criteria. Moreover, the mean age of first diagnosis of ADHD was 7.7 years in Taiwan [22], and the mean age of first diagnosis of ASD was 4 to 5 years [23], which were comparable to the results of an epidemiological study in the US [24]. The exclusion criteria were (1) missing information on birth year, (2) children less than 5 years old in 2016, and (3) twins, triplets, or other multiple births. If children were born in multiple births, we randomly selected one child from multiple births because genetic similarity is different between full siblings and monozygotic multiple births, and there is no information on monozygotic for multiple births in the TMCHD. If participants died before the end of the study (31 December 2017), they were treated as censored data by extracting their death date. The death status of participants is based on the Cause of Death Data, which was managed by the Office of Statistics, Department of Health, Taiwan. This study was approved by the Research Ethics Committee of the China Medical University and Hospital (approval number: CMUH108-REC1-142).

### 2.2. Measures

#### 2.2.1. Exposure

The exposure in the current study was the three allergic diseases (i.e., asthma, allergic rhinitis, or atopic dermatitis) and four autoimmune diseases (i.e., rheumatoid arthritis, Sjogren syndrome, psoriasis, and systemic lupus erythematosus) of children or their first-degree relatives (i.e., mother, father, and full sibling). The reason to choose these diseases was that these diseases are relatively common allergic and autoimmune diseases based on our preliminary analysis. If participants received one inpatient or at least two outpatient diagnoses of these autoimmune disease or allergic diseases during the research period (1 January 2004, to 31 December 2017), they were identified as having any allergic or autoimmune disease (Please see Table 1). These diagnoses were in accordance with the *International Classification of Disease, Ninth Revision* (ICD-9) and *Tenth Revision* (ICD-10), which were the diagnoses system of TMCHD and TNHIRD. The details are listed in Appendix A.

#### 2.2.2. Outcomes and Covariates

The main outcome evaluated in the current study was the occurrence of ASD and/or ADHD among children. We identified children as having ADHD or ASD if they received at least one inpatient diagnosis of ADHD or ASD (ICD-9 code: 299 or 314; ICD-10 code: F84 or F90, respectively) or more than two outpatient diagnoses given by psychiatrists within the study period. The covariates in this study were sociodemographic characteristics, namely parental age of delivery and child’s sex.

### 2.3. Statistical Analysis

To examine the associations of familial, siblings’ and children’s allergic and autoimmune diseases with children’s ASD and ADHD, we used the Cox regression model. In these models, children’s and familial (i.e., mother, father, or full sibling; those first-degree relatives averagely have 50% genetic similarity and half-sibling; second-degree relative averagely have 25% genetic similarity) allergic and autoimmune diseases were treated as predictors, whereas children’s ASD and ADHD were treated as the outcome. We adjusted for child and relative’s year of birth and familial clustering using the robust variance. We followed the analysis based on Hegvik et al. [25] to minimize residual confounding of age effect by modelling age effect as restricted cubic splines. The number and position of knots was based on the suggestion of Harrell that that the number of 4 knots offers an adequate fit of model for many datasets and the recommended quantiles for the position of a 4-knot model is 0.05, 0.35, 0.65, and 0.95 [26].

We used hazard ratios (HRs) with 95% confidence intervals (CIs) to quantify ASD or ADHD risks and their levels of statistical significance. An HR > 1 suggested that the risk of ASD or ADHD was higher in children in the comparison group than in the reference group. The CI of the including a null value of 1 implied that the risk of ASD or ADHD did not reach statistically significant difference.

We also examined whether higher number of affected relatives (i.e., within-individual, mother and father) in family associated with a higher incidence of ASD and/or ADHD, we participants into three groups: (1) no family member has allergic or autoimmune diseases, (2) only one family member (within-individual, mother or father) had allergic or autoimmune diseases, and (3) two family members had allergic or autoimmune diseases. We did not consider the group of within-individual, mother and father all had allergic or autoimmune diseases because such is an extremely rare case.

Finally, to further examine whether these familial associations of allergic and autoimmune diseases with ASD and/or ADHD were different between mothers and fathers. We used a moderation analysis with the inclusion of interaction between parents and exposure (i.e., allergic and autoimmune diseases), and the estimate of the interaction represents a differential effect between mothers and fathers [25]. In this analysis, children’s father with allergic or autoimmune disease was identified as reference group, and the HRs were the estimates of risk of ASD or ADHD for children’s mother with allergic or autoimmune disease.

## 3. Results

In total, we included 1,386,260 children in the current study. Among them, 661,957 (47.8%) children were boys and 724,303 children were girls (52.2%), and 78,307 children have at least one full sibling, with 351,192 of sibling pairs and 14,576 having one half-sibling. The average age of children was 9.7 years. The incidence of ASD and ADHD for these children was 1.0% and 5.1%, respectively. The demographics and disease status are presented in Table 1. The distribution of ADHD, ASD, immune disease, and autoimmune disease for half-sibling were presented in the Appendix A.

Table 2 shows the results of the Cox regression model examining the association between common autoimmune and allergic disorder and child’s ASD and ADHD using familiar aggregation. The association between ASD and most of the autoimmune diseases demonstrates insignificance among individuals, their fathers, mothers, and full siblings, whereas sibling’s SLE is significantly associated with children’s ASD (HR = 1.19). We found significant associations between children’s ASD and most of the allergic disease within individuals, their mother, father, and siblings. However, children’s ASD is not significantly associated with father’s atopic dermatitis, sibling’s asthma, allergic rhinitis, and atopic dermatitis. On the other hand, the association between individual’s as well as familial autoimmune disease and children’s ADHD is more pronounced than those with ASD. Among them, allergic diseases of mother are all significantly associated with children’s ADHD (HR = 1.13~1.49). Similarly, the associations between children’s ADHD and individual’s as well as familial allergic disease are pronounced. The details are listed in Table 2. Regarding the effect of half-sibling with allergic or autoimmune disease, only half-sibling with SLE is significantly associated with children’s ASD (HR = 1.48). The other details are listed in Appendix A.

In Table 3, we present the predominant effect of familial aggregation for allergic and autoimmune disease. Among children with ASD, most of the HRs are higher in the group comparison between two family members had allergic or autoimmune diseases and no family member has allergic or autoimmune diseases in comparison with the group comparison between only one family member had allergic or autoimmune diseases and no family member has allergic or autoimmune diseases. It demonstrates the effect familial aggregation, indicating that group with higher numbers of allergic or autoimmune diseases have higher HRs for children’s ASD. Similarly, this effect can be observed in the association between familial allergic or autoimmune disease and children’s ADHD. For allergic disease, the HRs all reach statistical significance among children’s ASD or ADHD.

Finally, the Table 4 demonstrates the risk of ASD or ADHD in the comparison between children’s father with allergic or autoimmune disease and children’s mother with allergic or autoimmune disease (father as reference). For development of children’s ASD, mother’s any of allergic disease (HR = 1.06), asthma (HR = 1.16) and allergic rhinitis (HR = 1.06) show the significantly higher risks than father’s disease. On the other hand, mother’s any of autoimmune disease (HR = 1.18), rheumatoid arthritis (HR = 1.42), any of allergic disease (HR = 1.20), asthma (HR = 1.23), allergic rhinitis (HR = 1.20), and atopic dermatitis (HR = 1.18) reveal significantly higher risk of ADHD among children in comparison with father’s disease.

## 4. Discussion

The current study demonstrated the significant association between familial allergic or autoimmune disease and development of ASD or ADHD among children. Furthermore, we also identified the predominant impact of familial aggregation on the associations between familial allergic/autoimmune diseases and development of ADHD or ASD among children. In advance, the associations between some parental diagnosis of autoimmune or allergic diseases in children’s ASD and/or ADHD was stronger in mothers than those in fathers, indicating the predominant effect with mothers. We found that any allergic disease from full sibling was significantly associated with children’s ASD and/or ADHD. However, the significance was not identified within individual disease. This may be due to the limited cases of groups with full sibling, and it will compromise the statistical power, resulting in the insignificance. Similarly, it can also explain the insignificance of adjusted odds ratio for two or three family members have disease in the column of ADHD in the Table 3.

The present study extends the application of the previous population-based study, demonstrating that ADHD and autoimmune diseases may co-aggregate among biological relatives, indicating that the relationship between ADHD and autoimmune diseases may be partially explained by shared genetic factors [25]. Our study further investigated the effect of familial aggregation on the association between familial allergic diseases and development of ADHD among children, as well as the association between familial autoimmune diseases and development of ASD. The results of previous studies on the effects of parental allergic and autoimmune diseases on children’s neurodevelopmental disorders have been mixed [16,18]. However, genetic evidence supports the association between autoimmune/allergic diseases and neurodevelopmental disorders [15]. Another epidemiological study implicated the association between the individuals with ADHD and their first-degree relatives’ autoimmune diseases [17]. The current study supported the propositions that familial allergic diseases are associated with the increased risks of ASD and ADHD in their children, as well as that some familial autoimmune diseases were associated with children’s ADHD. Although children may have no allergic or autoimmune diseases, their parents’ allergic and autoimmune diseases may increase the risk of ASD and ADHD in children.

Several reasons may explain the influence of parental immunologic diseases on the development of children’s neurodevelopmental disorders. First, allergic and autoimmune diseases have high heritability, implying an increased risk of allergic and autoimmune diseases in the children of parents with such diseases. A family history of immunological diseases is a key risk factor for allergic diseases [27]. Genetic and epigenetic factors play a prominent role in the emergence of autoimmune diseases [28]. Parental allergic and autoimmune diseases may, therefore, contribute to the occurrence of allergic and autoimmune diseases in children. Given that children’s allergic and autoimmune diseases and neurodevelopmental disorders may occur simultaneously [29,30], parental allergic and autoimmune diseases may increase the risks of ASD and ADHD in children.

Second, the derailment of the dopaminergic system may be associated with immune diseases. Dopamine receptors have been found to be expressed in immune cells, contributing to immune-inflammatory diseases [31]. Dysregulation of peripheral dopamine concentration has been reported to result in Parkinson’s disease, multiple sclerosis, rheumatoid arthritis, and inflammatory bowel disease [32]. Furthermore, an animal study revealed that maternal immune activation could disrupt the dopaminergic system in offspring [33]. With the association between dopaminergic dysregulation and ASD and ADHD [34,35], it is possible that the association between immune diseases and dopaminergic dysregulation may be related to the development of ASD and ADHD. On the other hand, the immune cells and cytokines of mothers may interact with the immunological system of the fetus. Maternal–fetal cellular trafficking, which is a physiological phenomenon involving the bidirectional passage of cells between mother and fetus during pregnancy, plays a key role in fetal immune system development [36]. One study identified the notable effect of maternal microchimerism on the priming of the fetal immune system in an animal model [37]. However, this physiological phenomenon may be associated with development of immune diseases in offspring. Murine fetuses that were exposed to microchimeric T cells from mothers and reactive to pancreatic beta cells exhibited an increased incidence of autoimmune diabetes compared with when the T cells were generated in the fetuses [38]. In addition, human studies have also explored the role of maternal microchimerism in the incidence of autoimmune diseases among children [30,31]. Maternal cells have been identified at higher rates in the cardiac specimens of patients with neonatal lupus syndrome [39] and in the blood as well as muscle tissues of patients with juvenile dermatomyositis compared with control groups [40]. These investigations support an etiology that involves maternal cells crossing the placental barrier and participating in the development of immune problems in children. Moreover, experimental studies have revealed that in utero exposures to maternal antibodies (e.g., immunoglobin G) and cytokines (interleukin-17) are key risk factors for ASD [41,42]. Consequently, cells and cytokines associated with maternal allergic or autoimmune diseases are postulated to pass the placental barrier to trigger the development of neurodevelopmental disorders. The present study discovered that maternal immune diseases had a stronger association with children’s ASD and/or ADHD compared with paternal ones. However, further biological and epidemiological studies are warranted to clarify the etiology and causality within this association.

Several limitations of this study should be noted. First, the nature of the retrospective cohort study limited the interpretation of the outcome due to the possibility of unmeasured variables. For instance, the effect of pharmacological treatment for allergic and autoimmune diseases on the development of neurodevelopmental diseases remains unknown. Second, the intertwined biomedical mechanisms of allergic and autoimmune diseases render subgroup analyses of either allergic or autoimmune disease influences individually difficult. Third, because the TMCHD was established since 2004 and by the end of this study (i.e., 2017), we can only include children born during this period. As a result, the age coverage of our child participants was limited to, at most, 13 years old. Furthermore, it is not easy to identify more than two generations in a dataset started in only 13 years. Therefore, our familial aggregation study design was limited to first-degree relatives but not to expand to second-degree or third-degree relatives. Fourth, due to the limitation of database, no further data of genetic information were available to clarify the association between familial autoimmune or allergic diseases and ASD or ADHD among children. Further genetic studies may be helpful to clarify the etiologies of this association. Finally, the TMCHD could not identify participants who did not seek medical help for ASD, ADHD, or allergic or autoimmune diseases. As TMCHD is a database with high coverage of the Taiwanese population [21], this effect may have been minimized. However, the prevalence of ASD and ADHD in TNHIRD were underestimated according to the comparison study between Taiwan’s National Epidemiological Study of Child Mental Disorders and TNHIRD [43]. The stigma [44] and gap in medical utilization for mental health between urban and rural areas [45] in Taiwan may contribute to the diversity.

## 5. Conclusions

The current study determined the influences of familial and children’s allergic and autoimmune diseases on the risks of ASD and/or ADHD among children. We also manifested the influence of familial aggregation in the association between parental allergic and autoimmune diseases on their children to develop ASD or ADHD. Physicians should pay attention to the possibility of comorbid ASD and ADHD among children with allergic or autoimmune diseases, especially among children with family histories of allergic or autoimmune diseases. Further study to investigate the biological mechanism underlying the association between immunological diseases, ASD, and/or ADHD is warranted.

## Figures and Tables

**Table 1 ijerph-19-04503-t001:** Demographics and autoimmune diseases and allergic diseases and child ADHD and/or ASD.

Variable	N = 1,386,260
*Child’s covariates*	
Sex	
Boys	661,957 (47.8)
Girls	724,303 (52.2)
Age	9.7 ± 2.3
Disease	
Autism spectrum disorder	13,885 (1.0)
Attention-deficit/hyperactivity disorder	71,219 (5.1)
Any allergic diseases	756,970 (54.6)
Asthma	354,301 (25.6)
Allergic rhinitis	565,663 (40.8)
Atopic dermatitis	276,898 (20.0)
Any autoimmune diseases	3487 (0.3)
Rheumatoid arthritis	420 (0.03)
Sjogren syndrome	544 (0.04)
Psoriasis	2140 (0.2)
Systemic lupus erythematosus	440 (0.03)
Ankylosing spondylitis	98 (0.01)
*Mother’s covariates*	
Age of childbirth	29.8 ± 4.7
Any allergic diseases	417,840 (30.1)
Asthma	88,399 (6.4)
Allergic rhinitis	355,629 (25.7)
Atopic dermatitis	56,810 (4.1)
Any autoimmune diseases	35,930 (2.6)
Rheumatoid arthritis	9074 (0.7)
Sjogren syndrome	12,841 (0.9)
Psoriasis	12,752 (0.9)
Systemic lupus erythematosus	5941 (0.4)
Ankylosing spondylitis	3722 (0.3)
*Father’s covariates*	
Age of childbirth	32.8 ± 5.3
Any allergic diseases	322,019 (23.2)
Asthma	67,629 (4.9)
Allergy rhinitis	280,981 (20.3)
Atopic dermatitis	25,265 (1.8)
Any autoimmune diseases	25,247 (1.8)
Rheumatoid arthritis	6364 (0.5)
Sjogren syndrome	4177 (0.3)
Psoriasis	14,729 (1.1)
Systemic lupus erythematosus	918 (0.1)
Ankylosing spondylitis	11,189 (0.8)

**Table 2 ijerph-19-04503-t002:** The association between common autoimmune and allergic disorder and child’s ASD and ADHD using familiar aggregation.

	Within-Individual		Mother		Father		Full-Sibling	
	aHR (95% CI)	*p*-Value	aHR (95% CI)	*p*-Value	aHR (95% CI)	*p*-Value	aHR (95% CI)	*p*-Value
*Autism spectrum disorder*								
Any autoimmune diseases	1.00 (0.71–1.40)	0.977	1.05 (0.96–1.16)	0.292	1.03 (0.92–1.17)	0.584	1.00 (0.60–1.66)	0.985
Rheumatoid arthritis	1.92 (0.96–3.86)	0.067	1.13 (0.94–1.37)	0.201	1.03 (0.82–1.31)	0.778	0.57 (0.08–4.04)	0.571
Sjogren syndrome	0.94 (0.40–2.23)	0.890	1.10 (0.94–1.30)	0.242	0.96 (0.70–1.31)	0.775	1.15 (0.64–2.07)	0.636
Psoriasis	0.91 (0.58–1.42)	0.666	0.96 (0.80–1.15)	0.660	1.01 (0.86–1.18)	0.904	1.09 (0.28–4.33)	0.899
SLE	0.68 (0.22–2.13)	0.508	1.14 (0.90–1.44)	0.277	0.95 (0.49–1.83)	0.868	1.19 (1.13–1.25)	<0.001
Any allergic diseases	1.67 (1.62–1.74)	<0.001	1.30 (1.25–1.36)	<0.001	1.20 (1.16–1.24)	<0.001	1.16 (1.09–1.24)	<0.001
Asthma	1.40 (1.35–1.45)	<0.001	1.38 (1.30–1.46)	<0.001	1.16 (1.08–1.25)	<0.001	0.99 (0.59–1.65)	0.969
Allergic rhinitis	1.79 (1.73–1.85)	<0.001	1.30 (1.26–1.34)	<0.001	1.20 (1.16–1.24)	<0.001	0.57 (0.08–4.04)	0.571
Atopic dermatitis	1.41 (1.36–1.46)	<0.001	1.12 (1.04–1.21)	0.002	1.03 (0.91–1.16)	0.640	1.16 (0.65–2.10)	0.612
*Attention-deficit/hyperactivity disorder*								
Any autoimmune diseases	1.12 (0.97–1.29)	0.123	1.26 (1.21–1.32)	<0.001	1.06 (1.00–1.13)	0.033	1.06 (0.86–1.30)	0.585
Rheumatoid arthritis	1.34 (0.92–1.94)	0.127	1.40 (1.29–1.51)	<0.001	1.00 (0.89–1.12)	0.933	1.29 (0.73–2.26)	0.380
Sjogren syndrome	0.67 (0.43–1.04)	0.071	1.30 (1.21–1.40)	<0.001	1.15 (1.01–1.31)	0.029	1.05 (0.80–1.37)	0.725
Psoriasis	1.28 (1.09–1.51)	0.002	1.13 (1.05–1.22)	<0.001	1.07 (1.00–1.14)	0.050	1.71 (1.04–2.82)	0.035
SLE	0.84 (0.54–1.30)	0.426	1.28 (1.16–1.41)	<0.001	1.09 (0.82–1.44)	0.553	1.26 (1.24–1.28)	<0.001
Any allergic diseases	1.65 (1.63–1.69)	<0.001	1.35 (1.33–1.37)	<0.001	1.12 (1.10–1.14)	<0.001	1.22 (1.19–1.25)	<0.001
Asthma	1.44 (1.42–1.46)	<0.001	1.49 (1.46–1.54)	<0.001	1.19 (1.15–1.23)	<0.001	1.06 (0.87–1.31)	0.553
Allergic rhinitis	1.74 (1.71–1.77)	<0.001	1.35 (1.32–1.38)	<0.001	1.11 (1.09–1.13)	<0.001	1.28 (0.73–2.25)	0.390
Atopic dermatitis	1.34 (1.31–1.37)	<0.001	1.21 (1.16–1.27)	<0.001	1.02 (0.97–1.07)	0.441	1.05 (0.81–1.38)	0.698

Systemic lupus erythematosus = SLE. Analyses were conducted with adjustment for individual and relative’s nonlinear age effects modelled by the restricted cubic spline function in the Cox regression.

**Table 3 ijerph-19-04503-t003:** The associations between the different number of family members (child, mother, and father) have common autoimmune and allergic disorder and child’s ASD and/or ADHD.

Disease	None Has This Disease	Only One Family Member Has This Disease	Two or Three Family Members Have This Disease	ASD				ADHD			
	n	n	n	aHR1 (95% CI)	*p*	aHR2 (95% CI)	*p*	aHR1 (95% CI)	*p*	aHR2 (95% CI)	*p*
Any autoimmune diseases	1,323,304	61,303	1653	1.02	0.635	1.45	0.070	1.17	<0.001	1.37	0.001
				(0.94–1.11)		(0.97–2.15)		(1.13–1.22)		(1.13–1.65)	
Rheumatoid arthritis	1,370,606	15,450	204	1.10	0.180	1.82	0.231	1.22	<0.001	1.28	0.378
				(0.96–1.28)		(0.68–4.86)		(1.14–1.31)		(0.74–2.22)	
Sjogren syndrome	1,369,102	16,769	389	1.01	0.845	1.91	0.071	1.25	<0.001	1.18	0.433
				(0.88–1.17)		(0.95–3.86)		(1.18–1.32)		(0.78–1.81)	
Psoriasis	1,357,064	28,798	398	0.98	0.739	1.25	0.615	1.11	<0.001	1.55	0.019
				(0.87–1.10)		(0.52–3.00)		(1.06–1.16)		(1.07–2.23)	
SLE	1,379,021	7180	59	1.08	0.457	1.65	0.623	1.24	<0.001	1.29	0.620
				(0.88–1.34)		(0.23–11.95)		(1.13–1.36)		(0.47–3.58)	
Any allergic diseases	423,086	534,230	428,944	1.40	<0.001	1.85	<0.001	1.44	<0.001	1.86	<0.001
				(1.34–1.47)		(1.77–1.93)		(1.41–1.47)		(1.82–1.90)	
Asthma	952,772	364,005	69,483	1.32	<0.001	1.64	<0.001	1.39	<0.001	1.76	<0.001
				(1.27–1.37)		(1.54–1.74)		(1.37–1.41)		(1.71–1.81)	
Allergic rhinitis	583,521	479,033	323,706	1.45	<0.001	1.87	<0.001	1.46	<0.001	1.83	<0.001
				(1.39–1.51)		(1.80–1.94)		(1.43–1.49)		(1.79–1.87)	
Atopic dermatitis	1,061,009	275,223	50,028	1.29	<0.001	1.51	<0.001	1.42	<0.001	1.87	<0.001
				(1.25–1.33)		(1.46–1.58)		(1.39–1.45)		(1.82–1.92)	

Systemic lupus erythematosus = SLE. Analyses were conducted with adjustment for individual and mother’s and father’s nonlinear age effects modelled by the restricted cubic spline function in the Cox regression. aHR1: adjusted odds ratio for only-one family member has this disease vs. no family member has this disease; aHR2: adjusted odds ratio for two or three family members have this disease vs. no family member has this disease.

**Table 4 ijerph-19-04503-t004:** The interaction by parent type on the association between autoimmune diseases or allergic diseases and autism spectrum disorder (ASD) or attention-deficit/hyperactivity disorder (ADHD).

Disease	ASD		ADHD	
	aHR (95% CI)	*p*-Value	aHR (95% CI)	*p*-Value
Any autoimmune diseases	1.02 (0.87–1.19)	0.808	1.18 (1.10–1.27)	<0.001
Rheumatoid arthritis	1.10 (0.82–1.50)	0.520	1.42 (1.24–1.63)	<0.001
Sjogren syndrome	1.15 (0.83–1.61)	0.401	1.13 (0.98–1.30)	0.093
Psoriasis	0.93 (0.73–1.18)	0.529	1.04 (0.95–1.15)	0.392
Systemic lupus erythematosus	1.20 (0.59–2.44)	0.612	1.16 (0.87–1.56)	0.315
Any allergic diseases	1.06 (1.01–1.12)	0.010	1.20 (1.17–1.23)	<0.001
Asthma	1.16 (1.06–1.27)	<0.001	1.23 (1.19–1.29)	<0.001
Allergic rhinitis	1.06 (1.01–1.11)	0.018	1.20 (1.18–1.22)	<0.001
Atopic dermatitis	1.02 (0.87–1.19)	0.808	1.18 (1.10–1.27)	<0.001

Identified father as reference group; analyses were conducted with adjustment for individual and mother’s and father’s nonlinear age effects modelled by the restricted cubic spline function in the Cox regression.

## Data Availability

The data presented in this study are available on request from the corresponding author.

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
