# Peer review of "Associations between Allergic and Autoimmune Diseases with Autism Spectrum Disorder and Attention-Deficit/Hyperactivity Disorder within Families: A Population-Based Cohort Study"

_ijerph, 2022, doi:10.3390/ijerph19084503_

Round 1

Reviewer 1 Report

The interesting study focusing on association between neurodevelopmental disorders and allergic/autoimmune diseases. As some aspects of the study design and method section are unclear for me, I am unable of assesing the scientific quality of results.

1. The Authors state that Taiwan medical database is proven to be a reliable data source. Nevertheless, I would appreciate more data regarding how the diagnoses coded in the database were established. Which classification system is used in Taiwan?

2. The Authors include into analysis "any autoimmune or allergic desease". How "autoimmune disease" and "allergic disease" were defined?

3. The Authors in the limitation section notice, that only people seeking medical attention due to their problems have their diagnosis included into database. I would appreciate some more discussion regarding this limitation. Are the data extracted from the database comparable with epidemiological estimates obtained from other studies?

4. In particular, the Authors define the treshold of 5 years of age (or older) in the database analysis. Indeed, developmental problems associated with infantile autism usually at age 5 are usually clearly visible. Nevertheless, ADHD and Asperger syndrome may be noticed later, in primary school. I, as well as probably many other readers,  do not know at what age children start school in Taiwan. What is the average age of ADHD and ASD diagnosis in Taiwan?

I recommend expanding and discussing the above issues.

Author Response

Reply to Comments from the Reviewers:

Reviewer #1:

The interesting study focusing on association between neurodevelopmental disorders and allergic/autoimmune diseases. As some aspects of the study design and method section are unclear for me, I am unable of assessing the scientific quality of results.

1. The Authors state that Taiwan medical database is proven to be a reliable data source. Nevertheless, I would appreciate more data regarding how the diagnoses coded in the database were established. Which classification system is used in Taiwan?

Response: Thanks for your questions, we have added additional sentence to elaborate this issue in the section 2.2.1 as “These diagnoses were in accordance with the International Classification of Disease, Ninth Revision (ICD-9) and Tenth Revision (ICD-10), which were the diagnoses system of TMCHD and TNHIRD.” Please refer to line 136-138.

  1. The Authors include into analysis "any autoimmune or allergic disease". How "autoimmune disease" and "allergic disease" were defined?

Response: Thanks for your question. In this study, “autoimmune diseases” represent four common autoimmune diseases, including rheumatoid arthritis, Sjogren syndrome, psoriasis, systemic lupus erythematosus, and ankylosing spondylitis; “allergic diseases” include asthma, allergic rhinitis, and atopic dermatitis. We revised the sentence as below to make it clear. Please refer to line 128-133.

“The exposure in the current study was the three allergic diseases (i.e., asthma, allergy rhinitis or atopic dermatitis) and four autoimmune diseases (i.e., rheumatoid arthritis, Sjogren syndrome, psoriasis and systemic lupus erythematosus) of children or their first-degree relatives (i.e., mother, father and full sibling). The reason to choose these diseases was that these diseases are relatively common allergic and autoimmune diseases based on our preliminary analysis.”

  1. The Authors in the limitation section notice, that only people seeking medical attention due to their problems have their diagnosis included into database. I would appreciate some more discussion regarding this limitation. Are the data extracted from the database comparable with epidemiological estimates obtained from other studies?

Response: Thanks for your question. Since allergic or autoimmune diseases are major illness with many of intolerable symptoms, it may be less likely to underestimate the prevalence of them. However, previous evidences demonstrated the underestimation of ADHD and ASD. We will discuss this issue in the section of Limitation as “Finally, the TMCHD could not identify participants who did not seek medical help for ASD, ADHD, or allergic or autoimmune diseases. As TMCHD is a database with high coverage of the Taiwanese population 1, this effect may have been minimized. However, the prevalence of ASD and ADHD in TNHIRD were underestimated according to the comparison study between Taiwan’s National Epidemiological Study of Child Mental Disorders and TNHIRD 2. The stigma 3 and gap in medical utilization for mental health between urban and rural areas 4 in Taiwan may contribute to the diversity.” Please refer to line 318-324.

  1. In particular, the Authors define the threshold of 5 years of age (or older) in the database analysis. Indeed, developmental problems associated with infantile autism usually at age 5 are usually clearly visible. Nevertheless, ADHD and Asperger syndrome may be noticed later, in primary school. I, as well as probably many other readers, do not know at what age children start school in Taiwan. What is the average age of ADHD and ASD diagnosis in Taiwan?

Response: Thanks for your question. We added the descriptions in the section 2.1 as “We recruited children who were 5 years old or older in 2016 because children aged 5 years or older may develop sufficient symptoms to permit reliable ASD and ADHD diagnoses, and every child had a minimum follow-up period of 1 year (to 2017). In Taiwan, children can enter the kindergartens since two years old; teachers have enough time to observe children’s behaviors for clarifying symptoms of ASD or ADHD. Therefore, it is suitable to set 5 years old as one of inclusion criteria. Moreover, the mean age of first diagnosis of ADHD was 7.7 years in Taiwan5 and the mean age of first diagnosis of ASD was 4 to 5 years6, which were comparable to the results of an epidemiological study in US7.” Please refer to line 107-115.

  1. I recommend expanding and discussing the above issues.

Response: As your suggestions, we discuss the above issues in the above descriptions.

Reference

  1. Li C-Y, Chen L-H, Chiou M-J, Liang F-W, Lu T-H. Set-up and future applications of the Taiwan Maternal and Child Health Database (TMCHD). Taiwan Gong Gong Wei Sheng Za Zhi 2016; 35: 209-220.
  2. Chen YL, Kuo RN, Gau SS. Burden of mental disorders in children in the general population and in health facilities: discrepancies in years lived with disability based on national prevalence estimates between populations receiving care or not. Eur Child Adolesc Psychiatry 2021.
  3. Zhuang XY, Wong DFK, Cheng CW, Pan SM. Mental health literacy, stigma and perception of causation of mental illness among Chinese people in Taiwan. Int J Soc Psychiatry 2017; 63: 498-507.
  4. Chiang CL, Chen PC, Huang LY et al. Time trends in first admission rates for schizophrenia and other psychotic disorders in Taiwan, 1998-2007: a 10-year population-based cohort study. Soc Psychiatry Psychiatr Epidemiol 2017; 52: 163-173.
  5. Wang, L.J.; Yang, K.C.; Lee, S.Y.; Yang, C.J.; Huang, T.S.; Lee, T.L.; Yuan, S.S.; Shyu, Y.C. Initiation and persistence of pharmacotherapy for youths with attention deficit hyperactivity disorder in Taiwan. PLoS One. 2016, 11, e0161061. doi: 10.1371/journal.pone.0161061.
  6. Hwang YS, Weng SF, Cho CY, Tsai WH. Higher prevalence of autism in Taiwanese children born prematurely: a nationwide population-based study. Res Dev Disabil 2013; 34: 2462-8.
  7. Fountain C, King MD, Bearman PS. Age of diagnosis for autism: individual and community factors across 10 birth cohorts. J Epidemiol Community Health 2011; 65: 503-10.

Reviewer 2 Report

In this report, the authors have tried to investigate the association of the development of ADHD and ASD in children with their parent's autoimmune diseases. The paper has a huge sample set, yet there are a few concerns if improved could significantly improve the manuscript.

Is there any information available for parents' genetic mutations or any genetic details available for allergy or any auto-immune disease ?

Did the authors look for any molecular differences in the parents with allergy and their children for ADHD and ASD ?

Survival data could be represented for the data. 

Did the authors look for mothers' allergy or their parents in connection with children's ADHD and ASD?

Author Response

Reply to Comments from the Reviewers:

Reviewer #2:

In this report, the authors have tried to investigate the association of the development of ADHD and ASD in children with their parent's autoimmune diseases. The paper has a huge sample set, yet there are a few concerns if improved could significantly improve the manuscript.

1. Is there any information available for parents' genetic mutations or any genetic details available for allergy or any auto-immune disease?

Response: Thanks for your questions. In the section of introduction and discussion, we have elaborated the possible mechanisms with genetic studies regarding the association between familial allergic/autoimmune diseases and ASD/ADHD among children. However, no further genetic data were available in the TNHIRD or TMCHD. We will add some sentences in the section of limitation as “Fourth, due to the limitation of database, no further data of genetic information were available to clarify the association between familial autoimmune or allergic diseases and ASD or ADHD among children. Further genetic studies may be helpful to clarify the etiologies of this association.” Please refer to line 314-318.

  1. Did the authors look for any molecular differences in the parents with allergy and their children for ADHD and ASD?

Response: Thanks for your questions. However, it is impossible to look for molecular difference in the current study due to the limitation of database. We will elaborate this issue in the section of limitation as “Fourth, due to the limitation of database, no further data of genetic information were available to clarify the association between familial autoimmune or allergic diseases and ASD or ADHD among children. Further genetic studies may be helpful to clarify the etiologies of this association.” Please refer to line 314-318.

  1. Survival data could be represented for the data.

Response: Thanks for your comment. We added the results of Cox regression into the revised manuscript. The results are presented in Tables 2 to 4. Please refer to line 189-240.

  1. Did the authors look for mothers' allergy or their parents in connection with children's ADHD and ASD?

Response: Yes, we presented the results of Cox regression model examining the association between common autoimmune and allergic disorders and child's ASD and ADHD using familiar aggregation, including mother, father, and full-sibling. Please refer to line 189-204.